# New Insights into the Interplay between Non-Coding RNAs and RNA-Binding Protein HnRNPK in Regulating Cellular Functions

**DOI:** 10.3390/cells8010062

**Published:** 2019-01-17

**Authors:** Yongjie Xu, Wei Wu, Qiu Han, Yaling Wang, Cencen Li, Pengpeng Zhang, Haixia Xu

**Affiliations:** College of Life Science, Xinyang Normal University, Xinyang 464000, China; yongjx81@126.com (Y.X.); wuwei95@126.com (W.W.); hanqiu96@163.com (Q.H.); yalingw97@163.com (Y.W.); licencen2009@126.com (C.L.)

**Keywords:** hnRNPK, lncRNAs, miRNA, gene regulation, cancer

## Abstract

The emerging data indicates that non-coding RNAs (ncRNAs) epresent more than the “junk sequences” of the genome. Both miRNAs and long non-coding RNAs (lncRNAs) are involved in fundamental biological processes, and their deregulation may lead to oncogenesis and other diseases. As an important RNA-binding protein (RBP), heterogeneous nuclear ribonucleoprotein K (hnRNPK) is known to regulate gene expression through the RNA-binding domain involved in various pathways, such as transcription, splicing, and translation. HnRNPK is a highly conserved gene that is abundantly expressed in mammalian cells. The interaction of hnRNPK and ncRNAs defines the novel way through which ncRNAs affect the expression of protein-coding genes and form autoregulatory feedback loops. This review summarizes the interactions of hnRNPK and ncRNAs in regulating gene expression at transcriptional and post-transcriptional levels or by changing the genomic structure, highlighting their involvement in carcinogenesis, glucose metabolism, stem cell differentiation, virus infection and other cellular functions. Drawing connections between such discoveries might provide novel targets to control the biological outputs of cells in response to different stimuli.

## 1. Introduction

In the human genome, less than 2% of the sequence has proved to be protein-coding genes—the sequence mainly comprises noncoding segments which were once considered “junk sequences” [1]. Recently, studies with ultra-high-throughput transcriptome sequencing and genome-wide technology have altered our perception of noncoding RNAs (ncRNAs) from ‘junk’ transcriptional products to functional regulatory molecules, and revealed that nearly two-thirds of the human genome is pervasively transcribed into ncRNAs, indicating that the majority of the human transcriptome consists of ncRNAs [2,3,4]. Non-coding transcripts can generally be divided into three major categories, namely, small non-coding RNAs (e.g., miRNAs, piRNAs and snoRNAs), long non-coding RNAs (lncRNAs) and circular RNAs (circRNAs) [5,6]. In quantity, ncRNAs outnumber protein-coding genes, which have been linked to a wide range of molecular biological processes including chromatin remodeling, transcription, post-transcriptional modifications and signal transduction. Among the ncRNAs, miRNAs belong to a distinct class of small ncRNAs that negatively regulate gene expression at the post-transcriptional level through interacting with the 3′UTR of specific target mRNA by either promoting mRNA decay or by dampening translation [7,8], while lncRNAs belong to functionally divergent groups of ncRNAs that regulate gene expression and other molecular functions through interacting with DNA, RNA or proteins [9]. Overall, ncRNAs represent a burgeoning field of biomedical research, and form a network of processes that spans the cell not only spatially as RNAs move across the cell, but also temporally as the RNAs regulate gene expression during cell cycle biological processes.

As one group of trans-acting factors, RNA-binding proteins (RBPs) are typically thought of as proteins that bind RNA through one or multiple RNA-binding domains (RBDs), such as the RNA recognition motif (RRM), the hnRNPK homology domain (KH) or the DEAD box helicase domain, and have been accepted as key regulators in transcriptional and post-transcriptional events [10]. RBPs with high affinity and/or specificity for their targets are more likely to have ascertainable biological functions that act to alter the fate or function of the RNAs [10,11]. Conventionally, RBPs were described as the “mRNA’s clothes”, which ensure that different mRNA regions including 5′UTR, 3′UTR and CDS are, at times, covered or exposed, thereby helping the mRNA to progress through the different stages of its life [12]. However, recent advances in studying the ncRNAs, such as lncRNAs and miRNAs, reveal the existence of complex RBP–ncRNA interactions that function in multiple biological processes such as epigenetic, transcriptional and post-transcriptional events [13,14,15]. Therefore, understanding the mechanism of action of RBP–ncRNA interactions is expected to contribute to creating new opportunities for advances in our understanding of human disease and even cancer, and also in order to provide a plethora of much-needed novel prognostic biomarker candidates or potential therapeutic targets.

Heterogeneous nuclear ribonucleoproteins (hnRNPs) are a large family of RBPs that are crucial for multiple aspects of RNA metabolism, while hnRNPK is the most extensively studied member of the hnRNP family involved in both physiological and pathological processes, such as spermatogenesis, nervous system and ovary development, erythroid differentiation, organogenesis, and carcinogenesis [16,17,18]. HnRNPK is abundantly expressed in various human cells and localizes in both the nucleus and cytoplasm, which preferentially recognizes poly-C sequences of target RNAs through its three repeats of K homology domains (KH1, KH2 and KH3) [19]. Previous research has demonstrated that hnRNPK can bind to the 3′UTR of target mRNAs including *c-Src*, *c-Myc*, *p21*, *eIF4E*, *r15-LOX*, *UCP2* that encode proteins involved in cell proliferation, apoptosis, and differentiation [18,20]. Recently, increasing evidence indicates that hnRNPK also interacts with many ncRNAs directly and plays a central role in many fundamental cellular processes. In this review, we discuss how the interplay between hnRNPK and ncRNAs regulates gene expression, chromatin structure and the location of lncRNAs, and contributes to cellular homeostasis, participating in tumorigenesis, DNA damage, glucose metabolism, stem cell differentiation, virus infection and other cellular functions. This review represents an effort to accumulate the current knowledge on the topic, highlighting the contribution of hnRNPK–ncRNA in physiological or pathogenetic conditions.

## 2. The Function of lncRNAs and hnRNPK Interaction

As mentioned above, lncRNAs are the biggest class of ncRNAs. They are greater than 200 nt in length and are emerging as the key regulators of many important biological processes impacting differentiation, development, and disease in every branch of life [21,22,23]. These functions stem from their versatile roles in regulating gene expression in transcription and post-transcriptional level via a number of complex mechanisms [22]. For example, some lncRNAs can function in either cis or trans by sequence complementarity with RNAs or DNAs, forming molecular frames and scaffolds for the assembly of macromolecular complexes, while others may act as a “molecular scaffold” by assembling protein complexes to mediate transcriptional or post-transcriptional activation/repression [15,24]. Interestingly, an increasing amount of evidence suggests that hnRNPK is a key interactor of lncRNAs participating in a variety of fundamental mechanisms in human health and disease (Table 1). In this section, we summarize the findings of hnRNPK–lncRNA interaction in fundamental cellular processes. The diversified mechanisms will be discussed, demonstrating the pleiotropic roles of hnRNPK–lncRNA interaction in regulating gene expression networks.

### 2.1. LncRNA Interacts with HnRNPK to Regulate Gene Transcription

#### 2.1.1. *LncRNA-p21*

*LncRNA-p21*, ~3.0 kb in length, has been identified as located in and transcribed from the cell-cycle regulator gene *p21* locus; it is a novel regulator of cell proliferation, apoptosis and DNA damage response, and plays a key role in development and diseases [50,51]. *LncRNA-p21* regulates gene expression by associating with the RNA-binding protein hnRNPK. Two groups have confirmed that the p53-inducible *lncRNA-p21* bound to hnRNPK to mediate gene repression in response to DNA damage [25,26,27]. However, the molecular mechanism was different. Dimitrova et al. used a conditional knockout mouse model to investigate the effects of *lncRNA-p21* deletion and indicated that *lncRNA-p21* deficiency could reduce *p21* expression and promoted the proliferation of mouse embryonic fibroblasts (MEFs) [25]. They demonstrated that *lncRNA-p21* functions in concert with hnRNPK to promote p53-mediated expression of neighboring gene, *p21*, predominantly in cis, and they support a model in which *lncRNA-p21* and hnRNPK act as transcriptional coactivators for local p53-mediated transcription. However, Huarte et al. did not detect any diminution in the expression of *p21* upon *lncRNA-p21* knockdown in MEFs using siRNA. They suggested that *lncRNA-p21* is a downstream effector of p53-dependent transcriptional response and is shown to mediate its function in trans by associating with hnRNPK; this interaction is needed for the proper localization of hnRNPK at loci of repressed genes and the regulation of p53-mediated apoptosis. A potential explanation for the discrepancy between the two studies is the different approaches used to knockdown *lincRNA-p21*. There could be a more complicated function mechanism for hnRNPK–*lncRNA-p21* interactions in apoptosis and DNA damage response, and additional experiments are needed to clarify this issue in the future.

*LincRNA-p21* also plays a role in regulating somatic reprogramming through interacting with hnRNPK [28]. Mechanistically, *lincRNA-p21* blocks reprogramming by forming a complex with hnRNPK and either H3K9 methyltransferase SETDB1 or DNA methyltransferase DNMT1. In combination, the two factors form repressive complexes containing either SETDB1 or DNMT1, preserving the high levels of H3K9me3 or CpG methylation at the pluripotency gene promoters. *Nanog* and *Lin28a* are two important pluripotency regulators. *LincRNA-p21* may block somatic cell reprogramming by CpG methylation at the promoters of a subset of genes (e.g., *Nanog*) and H3K9me3 maintenance at the promoters of another set of pluripotency genes (e.g., *Lin28a*), which are both repressed by hnRNPK-mediated transcriptional regulation. Moreover, *Lin28a* and *Nanog* are significantly upregulated by *hnRNPK* or *lincRNA-p21* knockdown. Therefore, *lincRNA-p21* exerts its role by reprogramming mainly through hnRNPK, which may constitute the essential functional unit in somatic cell reprogramming regulatory networks.

#### 2.1.2. *TUNA* and *Lncenc1*

LncRNAs are emerging as important players in many biological processes including embryonic development, stem cell self-renewal, and differentiation [52]. Recently, Rana et al. created an unbiased and genome-scale high throughput shRNA library targeting 1280 lncRNAs in the mouse genome, and identified 20 lncRNAs that were required for the maintenance of mouse embryonic stem cell (mESC) pluripotency [29]. One lncRNA, namely, Tcl1 upstream neuron associated (*TUNA*), shows remarkable sequence conservation in vertebrates; it is required for the maintenance of pluripotency and is specifically expressed in the central nervous system (CNS) of zebrafish, mice, and humans. Indeed, *TUNA* expression was increased when mESCs differentiated toward the neural lineages, and *TUNA* depletion inhibited the neural differentiation of ESCs. *TUNA* formed an RNA–multiprotein complex with hnRNPK and PTBP1 that was specifically enriched at the promoters of *Sox2*, *Nanog* and *Fgf4*, and activated these pluripotency genes directly [29]. Importantly, the depletion of several of these proteins phenocopied *TUNA* depletion. Thus, *TUNA* represents a lncRNA that is important for at least two cell states (ESC pluripotency and neural differentiation) and probably operates through multiple molecular mechanisms. In addition, *Lncenc1*, a highly abundant long noncoding RNA in nESC (naive embryonic stem cells), also interacts with hnRNPK and PTBP1, which regulate the transcription of glycolytic genes, thereby maintaining the self-renewal of nESCs [30]. Upon the depletion of *Lncenc1*, the expression of glycolysis-associated genes is significantly reduced and the glycolytic activity is substantially impaired, as indicated by more than 50% reduction in the levels of glucose consumption, lactate production, and the extracellular acidification rate. The functions of Lncenc1-mediated hnRNPK and PTBP1 recruitment in linking energy metabolism and the naive state of ESCs may enhance the understanding of the molecular basis underlying naive pluripotency. These findings suggest that hnRNPK–lncRNAs interactions may be key regulators of complexity in the networks controlling stem cell biology and disease pathophysiology. 

#### 2.1.3. *pancEts-1*

Neuroblastoma (NB) is a malignant tumor arising from primitive neural crests, and has a poor prognosis. There is a need to explore novel therapies [53]. Recent evidence has shown the crucial roles of lncRNAs in the tumorigenesis and aggressiveness of NB. Li et al. identified *pancEts-1* (Ets-1 promoter-associated noncoding RNA) as a novel 1395 nt lncRNA associated with the poor outcome of NB through the mining of a public microarray dataset [31]. *PancEts-1* is up-regulated in NB tissues and cell lines and promotes the growth, invasion, and metastasis of NB cells in vitro and in vivo. In addition, *PancEts-1* bound to the KH2 domain of hnRNPK to facilitate its physical interaction with β-catenin, while hnRNPK stabilized the β-catenin by inhibiting proteasome-mediated degradation, before increasing the nuclear translocation and activity of β-catenin, resulting in transcriptional alteration of the target genes associated with NB progression. The knockdown of hnRNPK abolished changes in the biological features of NB cells induced by *pancEts-1*, which indicates that the *pancEts-1*/hnRNPK/β-catenin axis may be of potential value as a novel therapeutic target for NB.

#### 2.1.4. *CASC11* and *lncRNA 91H*

Wnt/β-catenin signaling is known to regulate a broad range of cellular processes through regulating the ability of the multifunctional β-catenin protein, and its activation was involved in various human cancers [54]. lncRNA cancer susceptibility candidate 11 (*CASC11*), located in the chromosome 8q24 gene desert, ~2.1 kb upstream of c-Myc, is overexpressed in colorectal cancer (CRC) tissues [32]. Mechanically, *CASC11* enhances cell growth and metastasis by interacting with hnRNPK, which can promote the nuclear accumulation of β-catenin and activate the Wnt/β-catenin pathway to accelerate growth and metastasis in CRC cells. In addition, c-Myc, one of Wnt’s target genes, can directly bind to the promoter regions of *CASC11* and increase promoter histone acetylation to enhance *CASC11* expression. Therefore, *CASC11* is able to regulate metastatic potential in vitro and in vivo via the above pathway, and the interaction between hnRNPK and *CASC11* plays a pivotal role in the function of *CASC11* in CRC pathogenesis as an activity regulator of Wnt/β-catenin signaling. In addition, another lncRNA, named *lncRNA 91H*, which is located at the position of the H19/IGF2 locus, was also observed with remarkably elevated levels in the serum of CRC patients and cancer lines [33]. Furthermore, hnRNPK directly interacted with *lncRNA 91H* in CRC cells, and its expression level was closely related to *lncRNA 91H* expression. Thus, *lncRNA 91H* enhancing CRC metastasis by modifying hnRNPK expression might be an early plasma-based biomarker for CRC recurrence or metastasis. In summary, both *CASC11* and *lncRNA 91H* participate in the development and progression of CRC by associating with hnRNPK, while they may exert their actions in CRC pathogenesis through distinctly different signaling pathways. These findings might provide a foundation for the development of an early diagnostic and prognostic biomarker for CRC and the determination of innovative therapeutic strategies.

#### 2.1.5. *EWSAT1*

RNA-binding protein EWS-FLI1 (EWS and transcription factor FLI1) can both activate and repress genes that are critical for cell transformation [55]. *LncRNA-277*, also known as *EWSAT1* (Ewing sarcoma associated transcript 1), is induced by EWS-FLI1 in bone marrow-derived mesenchymal progenitor cells that were isolated from pediatric patients [34]. The examination of *EWSAT1* expression facilitates the development of Ewing sarcoma via the repression of target genes, revealing that it plays an important role in the pathogenesis of Ewing sarcoma. Using protein arrays and RNA immunoprecipitation (RIP) assays, the authors found that hnRNPK can interplay with *EWSAT1* specifically. Interestingly, there is a marked overlap in hnRNPK-repressed genes and those repressed by *EWSAT1*, suggesting that hnRNPK and *EWSAT1* act together to repress the expression of a subset of target genes in the context of Ewing sarcoma. Thus, *EWSAT1* may function, in part, by regulating the interaction of hnRNPK with target genes, although *EWSAT1* probably represses many genes through interactions with other yet to be identified proteins.

#### 2.1.6. *SLINKY*

Clear cell renal cell carcinomas (ccRCCs) are the most common histologic subtype with a broad range of clinical behavior, and prognostic biomarkers are needed to stratify patients for appropriate management [56]. Using variable next-generation sequencing data, Gong et al. recently identified several candidate lincRNAs that are prognostic in ccRCC and the top candidate, *SLINKY* (survival-predictive lincRNA in kidney cancer), is validated in an ethnically distinct dataset of ccRCC samples and provides prognostic information independent of the tumor stage and grade [35]. *SLINKY* appears to be a novel lncRNA that is upregulated across several malignancies and that drives aggressiveness in ccRCCs by regulating cancer cell proliferation. Knockdown *SLINKY* in ccRCC cell lines reduces cell proliferation, causes cell-cycle arrest, and alters gene expression programs related to cell growth and survival. Interestingly, *SLINKY* enhances cancer cell proliferation—most likely through its interaction with hnRNPK. Furthermore, hnRNPK deficiency reproduces the effects of *SLINKY* knockdown on cell proliferation and altered gene expression (both the upregulation and downregulation), so the possible recruitment of hnRNPK to gene loci might serve to either activate or repress the genes. Taken together, *SLINKY* appears to be a novel lncRNA that is upregulated across several malignancies and that drives aggressiveness in ccRCCs by regulating cancer cell proliferation through the recruitment of hnRNPK.

#### 2.1.7. *MYCLo-2*

C-Myc, as a transcription factor, not only regulates the transcription of many protein-coding genes but also noncoding RNAs such as miRNAs and lncRNAs, and the downstream genes are involved in various cellular processes including cell cycle, differentiation, cell growth, metabolism, cell adhesion, chromosome instability, and cell transformation [57]. C-Myc-regulated lncRNA MYCLos are involved in c-Myc functions, including cell cycle regulation and tumorigenesis. One example of upregulated lncRNA in colon cancer is the c-Myc-regulated lncRNA called *MYCLo-2*, which represses *p21* transcription by interacting with hnRNPK, and plays a crucial role in cancer transformation and tumorigenesis [36]. Loss of *MYCLo-2* function decreases colon inhibitory cancer cell transformation and tumorigenesis, indicating that *MYCLo-2* is a new target of cancer and c-Myc-associated pathogenesis.

#### 2.1.8. *LncRNA-OG*

Bone formation is a complex process consisting of intramembranous ossification and endochondral ossification; bone marrow-derived mesenchymal stem cells (BM–MSCs) are the main source of osteoblasts in vivo [58]. Recently, Tang et al. analyzed lncRNA expression profiles during BM–MSC osteogenesis, and further found that *lncRNA-OG* (osteogenesis-associated lncRNA) significantly promotes BM–MSC osteogenesis in vitro [37]. *LncRNA-OG* is mainly located in the nucleus of BM–MSCs and is upregulated during BM–MSC osteogenic differentiation, which regulates the activation of the BMP (bone morphogenetic protein) signaling pathway via interactions with hnRNPK. Surprisingly, hnRNPK also positively regulates *lncRNA-OG* transcriptional activity by promoting H3K27 acetylation of the *lncRNA-OG* promoter. Thus, this study revealed a novel lncRNA with a positive function on BM–MSC osteogenic differentiation and provides a new insight into the relationship between hnRNPK and lncRNA.

#### 2.1.9. *LncRNA-LBCS*

Cancer stem cells (CSCs) are responsible for the initiation, propagation, metastasis, drug resistance and relapse of cancers [59]. *SOX2*, as a master regulator that maintains stemness in embryonic stem cells as well as the self-renewal of CSCs in several malignancies, also plays a critical role in drug resistance to anticarcinogen (e.g., paclitaxel and gemcitabine) in several cancers [60,61]. Recently, Chen et al. reported that a novel lncRNA *lncRNA*–*LBCS* is significantly downregulated in bladder cancer stem cells (BCSCs) and cancer tissue, and suppresses self-renewal and chemoresistance of BCSCs in vitro and in vivo [38]. Interestingly, *lncRNA-LBCS* directly binds to the hnRNPK–EZH2 complex and guides it to the *SOX2* promoter and induces H3K27me3 to inhibit SOX2 expression, contributing to the attenuation of bladder cancer initiation and chemoresistance. Therefore, the *lncRNA*–*LBCS*/hnRNPK–EZH2/SOX2 regulatory axis acts as a tumor suppressor in bladder tumorigenesis and progression, and could be considered as a therapeutic target for clinical intervention in chemoresistant bladder cancer. 

#### 2.1.10. *SCAT7*

The signaling pathways controlling accurate DNA replication during the S-phase are critical for normal cell proliferation, since replication errors could result in abnormal cell proliferation [62]. To understand the functional role of lncRNAs in S-phase regulation, Ali et al. characterized the dynamic transcriptional changes across the S-phase in real time, and identified 1145 temporally expressed S-phase-enriched lncRNAs using nascent RNA capture sequencing [39]. Among these, 570 lncRNAs show significant differential expression in at least two cancer types across TCGA (The Cancer Genome Atlas) datasets. *SCAT7*, termed S-phase cancer-associated transcripts 7, is one of the differentially expressed S-phase-enriched lncRNAs and is elevated in multiple cancers. Mechanistic investigations on *SCAT7* in multiple cancer types using chromatin oligo-affinity precipitation (ChOP) and RNA immunoprecipitation (RIP) assays reveal that it interacts with the hnRNPK/YBX1 complex. Furthermore, the recruitment of the *SCAT7*/hnRNPK/YBX1 RNP complex at the promoter regions of *FGFR2* and *FGFR3* promotes transcriptional activation of FGF/FGFR and its downstream PI3K/AKT and MAPK pathways, leading to sustained cell proliferation and tumor development [39]. Taken together, the *SCAT7*/hnRNPK/YBX1 RNP complex plays a crucial role in carcinogenesis, and *SCAT7* may be used as a prognostic marker in risk assessments as well as potential therapeutic regimens in the treatment of cancer.

### 2.2. LncRNA and hnRNPK Interaction Controls mRNA Stability and Translation

LncRNAs control gene expression at not only the transcriptional but also the post-transcriptional level. In this section, we will discuss a few lncRNAs that regulate mRNA stability and translation. The aberrant expression of Wnt/β-catenin signaling enhances c-Myc and leads to tumorigenesis, especially in colon cancer. *MYU*, named c-Myc upregulated lncRNA, is generated from the opposite strand of the *VPS9D1* gene and is mainly localized in the cytoplasm [40]. *MYU*, as a direct target gene of the Wnt/c-Myc pathway, is upregulated in most colon cancers and required for the tumorigenicity of colon cancer. The overexpression of *MYU* in colon cancer cells significantly promotes its proliferation and tumorigenesis characteristics. Conversely, *MYU* knockout reduces the cellular CDK6 (cyclin-dependent kinase 6), mRNA and protein levels. Further investigation of lncRNA-mediated CDK6 regulation uncovered the significant role of hnRNPK and miR-16 in fine-tuning *CDK6* mRNA stability and translation. In human colorectal cancer cells, *MYU* interacts with hnRNPK to stabilize CDK6 expression and thereby promotes the G1-S transition of the cell cycle. Mechanistically, MYU, hnRNPK, and *CDK6* mRNA form a complex in which hnRNPK prevents the binding of miR-16 to *CDK6* mRNA 3′UTR, whereas the binding of miR-16 to *CDK6* 3′UTR results in decay of its mRNA in noncancerous cells [40]. In this condition, *MYU* renders specificity in regulating the stability of *CDK6* mRNA by determining its binding with hnRNPK or miR-16. In addition, one group has identified and characterized a novel lncRNA *linc00460*, which is a cytoplasmic transcript with upregulated expression in non-small cell lung cancer and associated with a poor prognosis for NSCLC (Non-small cell lung cancer) patients, implying that *linc00460* is important for lung cancer development [41]. *Linc00460* can also physically and specifically interact with hnRNPK. In lung cancer, elevated *linc00460* binds with and translocates hnRNPK to the cytoplasm to participate in special mRNA stability and translation regulation, and also plays a key role in cell migration and invasion. However, the detailed molecular mechanism needs further investigation. Furthermore, Shin et al. reported that a lncRNA named *PTOV1-AS1* (prostate tumor overexpressed 1 antisense transcript 1), which directly interacts with hnRNPK and is directly regulated by hnRNPK, regulates the proto-oncogene heme oxygenase 1 (*HMOX1*) [42]. Both *PTOV1-AS1* and *HMOX1* mRNA have the same miRNA response element (MRE) of miR-1270-5p. Therefore, hnRNPK-mediated PTOV1-AS1 regulation acts as a decoy for miR-1207-5p to modulate *HMOX1* expression, which plays a critical role in the proliferation and metastasis of various cancers. 

Unlike lncRNA *MY*, *linc00460* and *PTOV1-AS1*, which promote the translation of target mRNAs, *treRNA* (translational regulatory lncRNA) (originally named as ncRNA-a7) reduces the level of E-cadherin protein (EMT) without affecting the levels of its mRNA [43]. *TreRNA* is increased in metastatic breast cancer and controls the metastatic potential by regulating a subset of EMT markers. Nevertheless, there is no sequence homology between *treRNA* and the 3′UTR of *E-cadherin* mRNA. Mechanically, *treRNA* interacts with hnRNPK and four other RNA-binding proteins (FXR1, FXR2, PUF60, and SF3B3) promoting the formation of a *treRNA*-associated protein (treRNP) complex, which interacts, directly or indirectly, with the 3′UTR of *E-cadherin* mRNA and reduces the translation efficiency. Altogether, these findings emphasize the role of lncRNAs acting as molecular anchors that fine-tune cell signaling by recruiting hnRNPK, resulting in tumorigenesis by the repression or activation of specific genes.

### 2.3. LncRNA and hnRNPK Interaction Promotes lncRNAs Nuclear Enrichment

LncRNAs are more commonly localized in the nuclear fraction, whereas mRNAs are usually exported to and enriched in the cytoplasm, although several reports have described that a small number of mRNAs still exist in the nucleus [63,64]. However, the mechanisms of lncRNAs’ and mRNAs’ nuclear enrichment remain unclear. A recent study demonstrated that short consensus sequences encoded in both lncRNAs and mRNAs are associated with the accumulation of RNAs in the nucleus [65]. These sequences shared a 42-nt fragment that contained three stretches of at least six pyrimidines (C/T), two of which were similar to each other and matched the consensus RCCTCCC (where R denotes A/G), and were found to overlap in antisense orientation with Alu repeat sequences—the most frequent short interspersed elements (SINEs) in the human genome. The authors named them SINE-derived nuclear RNA localization (SIRLOIN), which was also enriched in nuclear fractions in the public RNA sequencing datasets of ENCODE cell lines, suggesting that SIRLOINs may represent a universal mechanism of nuclear localization. Furthermore, the authors revealed the RNA-binding protein hnRNPK as the highest-ranking interaction partner with SIRLOIN elements, and confirmed this interaction by RIP assay of hnRNPK with SIRLOIN-supplemented *GFP* mRNA in MCF7 cells. Meanwhile, upon the siRNA-mediated knockdown of hnRNPK in MCF7 and HeLa cells, subcellular localization of lncRNAs was disturbed; specifically, SIRLOIN-containing RNAs showed decreased nuclear localization, which suggests that hnRNPK binding drives nuclear enrichment mediated by SIRLOIN elements [65]. Overall, this study suggests that SIRLOIN-containing RNAs are localized to the nucleus through their association with hnRNPK. The nuclear enrichment mechanism is more common in lncRNAs and some mRNAs, but the exact mechanism through which hnRNPK achieves the nuclear enrichment of SIRLOIN-containing RNAs still remains to be further elucidated.

### 2.4. LncRNA and hnRNPK Interaction Regulates Genome Structure

LncRNAs also play a central role in regulating chromatin structure and nuclear architecture [51,66]. The X inactive specific transcript (*Xist*) gene, as a 17 kb lncRNA, was first identified as a candidate for the master switch locus required in cis for the process of X chromosome inactivation, which can organize a repressive chromosome compartment as a physical scaffold [67,68,69]. HnRNPK is found as a major component of nuclear matrix preparations scaffold-attachment region-associated protein, consistent with the role in *Xist*-mediated silencing [70]. Several recent studies have reported that hnRNPK may act as a docking platform or hub for several complexes that mediate nucleic acid linked processes by interaction with *Xist* [44,45,46,47]. Chu et al. support a direct role for hnRNPK in *Xist*-mediated silencing in ESCs by assaying the silencing of genes located in cis to an inducible *Xist* transgene [45]. The knockdown of hnRNPK resulted in the detachment of *Xist* from the inactive X-chromosome, suggesting that hnRNPK is required for *Xist* RNA localization on the inactive X-chromosome. For further confirmation that hnRNPK interacts directly with *Xist* RNA, Cirillo et al. define a strong peak precisely corresponding to the cytidine-rich B-repeat of *Xist* RNA using hnRNPK iCLIP (individual-nucleotide resolution UV crosslinking and immunoprecipitation) analysis [46]. Then, Pintacuda et al. also suggest that tandemly arranged copies of the cytidine-rich B-repeat element recruit multiple hnRNPK subunits, and further revealed the molecular mechanism of hnRNPK in *Xist*-mediated chromatin modifications [44]. Moreover, hnRNPK, through its KI domain, interacts directly with the PCGF3/5 subunit of the PCGF3/5-PRC1 complex, which initiates the Polycomb cascade. Therefore, the RNA-binding protein hnRNPK bridges the B-repeat with the initiating Polycomb complex PCGF3/5-PRC1 to modify underlying chromatin, or alternatively it may dissociate and diffuse to nearby sites. In addition, Smchd1, as a non-canonical structural maintenance of chromosomes (SMC) family protein, is critically involved in both random and imprinted X chromosome inactivation during development, which also depends on the *Xist*/hnRNPK/PRC1 pathway for recruitment to the inactive X [47]. Accordingly, synthetically tethering hnRNPK to *Xist* RNA-lacking B-repeat is sufficient for *Xist*-dependent Polycomb recruitment. Given the potential importance of the interaction between lncRNA *Xist* and hnRNPK in guiding chromatin modification, this mechanism provides a model for further studies on lncRNA function both in development and in disease.

### 2.5. hnRNPK Regulates the Alternative Splicing of lncRNAs

Paraspeckles are unique lncRNA-directed nuclear substructures that are usually detected in cultured cell lines as a variable number of foci found in close proximity to the nuclear speckles, which play an important role in the regulation of gene expression through various mechanisms including mRNA retaining, mRNA breakage, A-to-I editing and protein seizing [48,71,72]. Paraspeckles were initially defined as foci comprised of characteristic RNA-binding proteins and specific lncRNAs. *Nuclear Enriched Abundant Transcript 1 (NEAT1*), a lncRNA with five known splice variants, is an essential structural constituent of paraspeckle nuclear bodies [73]. 3.7-kb *NEAT1_1* and 23.7-kb *NEAT1_2* are the two most abundant isoforms of *NEAT1* that are produced by alternative 3′-end processing. The short-isoform *NEAT1_1* is canonically polyadenylated, whereas the long-isoform *NEAT1_2* is stabilized by a triple helix structure processed by RNase P cleavage at its 3′ end. Interestingly, the production of the essential *NEAT1_2* isoform is regulated by hnRNPK, which can bind to the short pyrimidine stretch located between the canonical polyadenylation signal for *NEAT1_1* and the upstream CPSF6/NUDT21 (CFIm) complex binding cluster and interferes with *NEAT1_1* 3′-end processing, resulting in the preferential synthesis of *NEAT1_2* [48,49]. Mechanically, hnRNPK arrested binding of the CFIm complex in the vicinity of the alternative polyadenylation site of *NEAT1_1*, leading to the preferential accumulation of *NEAT1_2* and initiated paraspeckle construction. Furthermore, hnRNPK is also one of the essential paraspeckle proteins through binding *NEAT1_2* [48]. Thus, the hnRNPK-dependent regulatory mechanism in *NEAT1* 3′-end processing is required for *NEAT1_2* accumulation and the next paraspeckle formation. Further studies designed in order to understand the *NEAT1*–hnRNPK interaction and its dynamics could aid in understanding of the molecular mechanism underlying the dynamic nature of paraspeckles under normal or disease conditions.

## 3. Interplay between hnRNPK and miRNAs

MiRNAs are a class of endogenous, noncoding single-stranded ~22 nt RNA, which are evolutionarily conserved and contribute to the control of cellular processes such as proliferation, differentiation, growth, death, inflammation, and development [7,8,13]. Based on sequence complementarities, miRNAs target mRNAs 3′UTR by binding regions to “seed” sequences at miRNA 5′-termini (2–8 nt) and regulate posttranscriptional gene expression through transcript degradation or translation repression, or both. Recently, a number of reports have demonstrated that RBPs also interact with miRNAs through their RNA-binding motifs (e.g., RRM, KH, Pumilio, and SAM), and the interplay between specific RBPs and miRNAs may function in regulating the translation of multiple genes [13,52,74]. The interplay can be antagonistic, where the binding of the RBP prevents the binding of the target mRNA. Among the RBPs in the RBP–miRNA interaction, HuR (Human antigen R) is one of the most-studied RBPs, which can interact with a number of miRNAs (e.g., miR-548c, miR-494, miR-21, miR-16, miR-331-3p, miR-1192, miR-1285, miR-122, and miR-125b) to regulate the expression of miRNAs targeting mRNAs encoding cytokine receptors, oncogenes, tumor suppressors, and cell cycle regulators [75,76,77]. As an important RBP, hnRNPK is also known to regulate the expression of target genes involved in the hnRNPK–miRNA interaction. HnRNP K is upregulated in CML-BCcells, and inhibits myeloid cell differentiation via binding to a C-rich motif in the 5′UTR of *C/EBP*α, which is a master regulator of myeloid cell differentiation and survival [78,79]. Interestingly, miR-328 can interact with hnRNPK in a seed sequence-independent manner by acting as a decoy of hnRNPK, thus relieving translational inhibition of *C/EBP*α myeloid differentiation [78]. In addition, MiR-122 is a highly abundant RNA in hepatocytes that is required for the replication of the hepatitis C virus (HCV) [80,81]. Fan et al. identified that hnRNPK was able to bind to miR-122 through short pyrimidine sequences in miR-122 using human proteome chip screening, and the interaction between hnRNPK and miR122 will likely modulate miR-122 stability and the replication of HCV RNA [82]. Thus, it is likely that the decoy activity of miRNAs is not limited to miR-328 or miR-122 but can be extended to other miRNAs containing C-rich elements resembling the consensus RNA-binding sites for hnRNPK that are involved in different normal cell functions and in neoplastic as well as non-cancer-related diseases. More research is needed to discover hnRNPK–miRNAs interactions in the future.

## 4. Conclusions and Perspectives

NcRNAs, such as lncRNAs and miRNAs, are involved in cellular homeostasis affecting cell proliferation, cell growth, differentiation and apoptosis, while their disruption leads to malignant transformation of the cells. HnRNPK is a highly conserved gene that is abundantly expressed in mammalian cells. The interaction of hnRNPK and ncRNAs defines the novel way through which ncRNAs affect chromatin structure, the expression of protein-coding genes, and modulate pathways contributing to development, differentiation and oncogenesis. Both lncRNAs and miRNAs physically interact with hnRNPK and this leads to the regulation of RNA stability or translation. In addition, lncRNAs and miRNAs are able to cross-talk with hnRNPK to change the expression of their target genes competitively or cooperatively. Furthermore, various cancer-associated hnRNPK–lncRNAs axes function to regulate the cancer-associated phenotype through modulating target genes. In this review, we summarized the role of ncRNAs and hnRNPK interactions in regulating gene expression and cellular functions and emphasized gene regulation via the interaction of hnRNPK with ncRNAs, such as lncRNAs and miRNAs, at transcriptional and post-transcriptional levels. The mechanisms of action can mainly be categorized into the following groups: (1) lncRNAs recruit hnRNPK and turn on or off gene transcription by binding to gene promoters; (2) lncRNAs and hnRNPK interaction promotes lncRNAs nuclear enrichment; (3) lncRNAs, hnRNPK and mRNA form a complex that controls mRNA stability or translation; (4) hnRNPK acts as an endogenous decoy for miRNAs, affecting their distribution on their target genes; (5) lncRNAs regulate genome organization by interacting with hnRNPK, indirectly affecting gene expression; and (6) hnRNPK regulate the alternative splicing of lncRNAs (Figure 1).

Despite the progress in the ncRNAs field in recent years, research into the hnRNPK–ncRNAs interaction’s involvement in biological processes and human diseases is yet to be elucidated. To better understand the hnRNPK–ncRNAs regulatory network, there are several areas that warrant further investigation. Firstly, there is an urgent need for the systematical identification of hnRNPK-interplayed ncRNAs including lncRNAs, miRNAs and emerging circRNAs. Current research is still scattered, and a more systematic screen may be able to provide a more comprehensive picture of hnRNPK–ncRNAs interaction. To detect this possibility, a global search by RIP-seq with the hnRNPK antibody under various conditions, including physiological or pathogenetic conditions, would be one good approach. Secondly, the regulation of the hnRNPK–ncRNA interaction should be further investigated. The location and magnitude of the interaction for a specific ncRNA and hnRNPK must be associated to regulate the change of target gene expression and cellular function. For instance, the mechanisms that regulate the nucleo-cytoplasmic shuttling of hnRNPK and ncRNA complexes are important to investigate. Thirdly, Statello et al. recently discovered interplay between RNA-binding proteins and RNAs by packaging into exosomes, which could be important for the transport of RNAs into exosomes and the maintenance of RNAs inside exosomes [83]. These studies provide a new direction for studying hnRNPK and RNAs interactions in exosomes. Finally, given the important role of hnRNPK in cancer development, future research may also need to evaluate the utility of hnRNPK-associated ncRNAs as cancer biomarkers or novel therapeutic targets. For instance, targeting the ncRNAs associated with cell cycle regulation or apoptosis, along with conventional chemotherapy, may increase cancer therapy efficacy.

## Figures and Tables

**Figure 1 cells-08-00062-f001:**
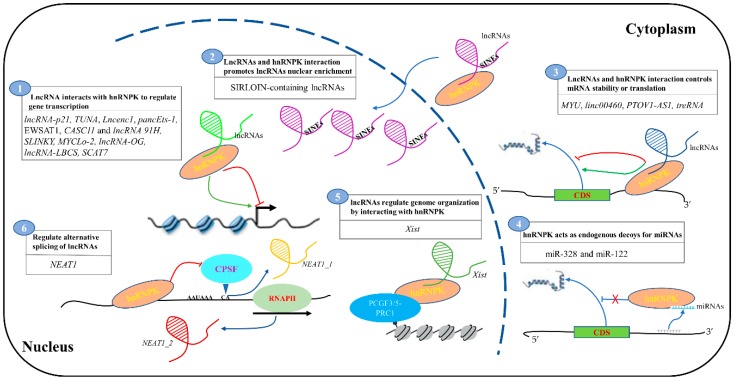
The molecular mechanisms of ncRNAs and hnRNPK interplay in regulating gene expression. (1) long non-coding RNAs (lncRNAs) recruit hnRNPK and promote or restrain gene transcription; for example, *Lncenc1*1 binds to hnRNPK and induces glycolysis-associated genes in maintaining the self-renewal of nESCs. (2) lncRNAs and hnRNPK interaction promotes SINE-derived nuclear RNA localization (SIRLOIN)-containing lncRNAs nuclear enrichment. (3) lncRNAs, hnRNPK and mRNA form a complex that controls mRNA stability or translation; for example, the *MYU* and hnRNPK complex regulates *CDK6* mRNA translation. (4) hnRNPK act as miRNAs sponges, affecting their distribution on their target genes; for example, hnRNPK interact with miR-328 by acting as a decoy, thus relieving translational inhibition of C/EBPα. (5) lncRNA *Xist* regulates genome organization by interacting with hnRNPK, indirectly affecting gene expression. (6) hnRNPK is involved in the alternative splicing of lncRNAs; for example, hnRNPK regulates alternative 3′-end processing of *NEAT1*, and is required to maintain the *NEAT1_2* level.

**Table 1 cells-08-00062-t001:** Molecular function of heterogeneous nuclear ribonucleoprotein K (hnRNPK) and long non-coding RNAs (lncRNAs) interaction.

lncRNAs	Location	Dysregulation	Function	Mechanism	Ref.
*LincRNA-p21*	Nucleus	Upregulated	Promote mouse MEFs proliferation, p53-mediated apoptosis, regulate somatic reprogramming	Transcriptional regulation	[25,26,27,28]
*TUNA*	Nucleus	Upregulated	ESC pluripotency and neural differentiation	activate the pluripotency genes	[29]
*Lncenc1*	Nucleus	Upregulated	Maintain the self-renewal of nESCs	Regulate the transcription of glycolytic genes	[30]
*pancEts-1*	Nucleus	Upregulated	Promote the growth, invasion, and metastasis of NB cells	Activate β-catenin	[31]
*CASC11*	Nucleus	Upregulated	Promote CRC cell proliferation and metastasis	Activate Wnt/β-catenin pathway	[32]
*lncRNA 91H*	Exosom	Upregulated	Promote colorectal cancer development and metastasis.	Regulate hnRNPK expression	[33]
*EWSAT1*	Nucleus	Upregulated	Facilitates the development of Ewing sarcoma	Repress the expression of a subset of target genes in the context of Ewing sarcoma	[34]
*SLINKY*	Nucleus	Upregulated	Regulate cancer cell proliferation	Transcriptional regulation	[35]
*MYCLo-2*	Nucleus	Upregulated	Colon cancer transformation and tumorigenesis	Repress *p21* transcription	[36]
*LncRNA-OG*	Nucleus	Upregulated	Promote BM-MSC osteogenic differentiation	Promoting H3K27 acetylation of the lncRNA-OG promoter	[37]
*LncRNA-LBCS*	Nucleus	Downregulated	Inhibits self-renewal, chemoresistance and tumor initiation of BCSCs	Repress SOX2 transcription via mediating H3K27me3	[38]
*SCAT7*	Nucleus	Upregulated	Promote cell proliferation and tumor development	Activate FGF/FGFR transcription and its downstream PI3K/AKT and MAPK pathways	[39]
*MYU*	Cytoplasm	Upregulated	Promote proliferation and tumorigenicity	Stabilize CDK6 expression	[40]
*linc00460*	Cytoplasm	Upregulated	Lung cancer development	Translocate hnRNPK to the cytoplasm	[41]
*PTOV1-AS1*	Cytoplasm	Upregulated	Promote proliferation and metastasis of various cancer	Modulate HMOX1 expression	[42]
*treRNA*	Cytoplasm	Upregulated	Promote tumorigenesis	Suppress translation of E-cadherin	[43]
*Xist*	Nucleus	Upregulated	Inactive X-chromosome	Modify underlying chromatin	[44,45,46,47]
*NEAT1*	Nucleus	-	Modulates the alternative *NEAT1* 3′-end processing	HnRNPK competed with CPSF6 for binding to NUDT21	[48,49]

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
