# Peer review of "New Insights into the Interplay between Non-Coding RNAs and RNA-Binding Protein HnRNPK in Regulating Cellular Functions"

_cells, 2019, doi:10.3390/cells8010062_

Round 1

Reviewer 1 Report

This review is well described and seems to be beneficial for researchers involved in this area. However, it seems to be difficult for researchers that are unfamiliar with this area to grasp the relationship between ncRNA and hnRNPK, because there is no schematic figure. The authors should add at least one schematic figure to this review for better understanding.

Recently, lncRNA NEAT1 is identified as a direct p53 target gene (Idogawa, et al. Int J Cancer 2017, Adriaens, et al. Nature Med 2016) and the role of NEAT1 in cancer and other diseases is received a great deal of attention. It is reported that hnRNPK interacts with NEAT1 and regulate NEAT1 function (Naganuma, et al. EMBO J 2012). The relationship between NEAT1 and hnRNPK should also be reviewed briefly.

Author Response

Reviewer 1

1. This review is well described and seems to be beneficial for researchers involved in this area. However, it seems to be difficult for researchers that are unfamiliar with this area to grasp the relationship between ncRNA and hnRNPK, because there is no schematic figure. The authors should add at least one schematic figure to this review for better understanding.

Response: Thanks! We have added a schematic figure in the reviewed manuscript.

2. Recently, lncRNA NEAT1 is identified as a direct p53 target gene (Idogawa, et al. Int J Cancer 2017, Adriaens, et al. Nature Med 2016) and the role of NEAT1 in cancer and other diseases is received a great deal of attention. It is reported that hnRNPK interacts with NEAT1 and regulate NEAT1 function (Naganuma, et al. EMBO J 2012). The relationship between NEAT1 and hnRNPK should also be reviewed briefly.

Response: Thanks! LncRNA NEAT1 is a famous ncRNA with multifunction in the regulation of gene expression, which also interacts with hnRNPK. We have discussed the relationship between NEAT1 and hnRNPK in the reviewed manuscript.

Reviewer 2 Report

In this article, authors discuss the regulatory interplays between non-coding RNAs and RNA binding protein hnRNPK

The review is well written, and the description of selected topic is well elaborated. While this review could be of general interest to readers, the authors may wish to revise their manuscript for following comments;

Central to current discussion is the hnRPK (RNA-binding protein), which binds with long ncRNAs and regulates the cellular functions. It would be imperative to provide the sequence of motif which binds with RNAs, as well as the domain structure. Authors may wish to provide small table showing the sequence (or sequences) of the motif which have been already reported to bind with RNA species. And provide a schematic structure of the domain(s).

While authors introduce the regulatory functions of long ncRNAs, I would suggest authors to refer the following article, from MDPI journal non-coding RNA (PMID: 29657282). Moreover, it would be worth mentioning newly discovered interplays between RNA-binding proteins and ncRNAs by packaging into small vesicles (exosomes), and their transport/shuttle to other cells, as has been reported by Statello et al, 2018 (PMID: 29689087).

Title: in biological context, I perceive the same meaning of role and interplay, therefore, in the title of manuscript authors may wish to remove either the word ‘’role’’ or ‘’interplay’’.

Author Response

1. Central to current discussion is the hnRNPK (RNA-binding protein), which binds with long ncRNAs and regulates the cellular functions. It would be imperative to provide the sequence of motif which binds with RNAs, as well as the domain structure. Authors may wish to provide small table showing the sequence (or sequences) of the motif which have been already reported to bind with RNA species. And provide a schematic structure of the domain(s).

Response: Thanks! HnRNPK preferentially recognizes poly-C sequences of target RNAs through its three repeats of K homology domains (KH1, KH2 and KH3). Bomsztyk et al have reviewed it (Bomsztyk et al 2004, reference 19). We have mentioned it in the introduction of manuscript. The most research did not provide the detailed sequence of the motif which band to hnRNPK, so we are not list the sequence using a table.

2. While authors introduce the regulatory functions of long ncRNAs, I would suggest authors to refer the following article, from MDPI journal non-coding RNA (PMID: 29657282). Moreover, it would be worth mentioning newly discovered interplays between RNA-binding proteins and ncRNAs by packaging into small vesicles (exosomes), and their transport/shuttle to other cells, as has been reported by Statello et al, 2018 (PMID: 29689087).

Response: Thanks! We have read this paper carefully and referred these inthe reviewed manuscript.

3. Title: in biological context, I perceive the same meaning of role and interplay, therefore, in the title of manuscript authors may wish to remove either the word “role” or “interplay’’.

Response: Thanks! We have corrected it in the reviewed manuscript.